# Exploring the Interior Designers' Attitudes toward Sustainable Interior Design Practices: The Case of Jordan

Mousa S. Mohsen [1],* and Rana Matarneh [2]

[1] Commission for Academic Accreditation, Ministry of Education, Abu Dhabi P.O.Box 295, United Arab Emirates
[2] Department of Architecture and Design, College of Engineering, Khawarizmi University Technical College, Amman 11953, Jordan; r.matarneh@khawarizmi.edu.jo
* Correspondence: mousa.mohsen@moe.gov.ae

**Abstract:** Interior designers play a pivotal role in shaping the built environment and catalyzing positive change through the adoption of sustainable design practices. This study centers on the analysis of prevailing attitudes held by interior designers in Jordan towards sustainable interior design practices. Through a comprehensive methodology involving a literature review and a three-part questionnaire, the research explores the benefits, challenges, and integration of sustainability principles. The study highlights substantial awareness (85%) of sustainable interior design's environmental impact and resource preservation. Additionally, 81% acknowledge its enduring significance and 89% recognize its diverse benefits. While 61% exhibit familiarity and 27% claim professional experience. Among sustainability indicators, energy efficiency scores 4.25, resource efficiency 4.27, and "Design aligns with laws and regulations by the Ministry of Labor" ranks highest at 4.37. This study significantly augments our understanding of sustainable interior design by introducing a comprehensive evaluation framework that encompasses the three sustainability dimensions. It equips decision makers with a robust tool to identify specific practices that bolster sustainability performance, further advancing the field. Furthermore, this study provides practical strategies for the application of sustainable interior design principles in the Jordanian context, emphasizing the need for hands-on training, interdisciplinary collaboration, policy development, and ongoing monitoring. These strategies aim to bridge the gap between awareness and practical experience, fostering a culture of sustainability within the interior design profession. The findings of this research resonate with existing literature on sustainability in the built environment, demonstrating a global shift towards sustainability as a fundamental approach rather than a passing trend. The introduced comprehensive evaluation framework equips decision makers with a robust tool to identify specific practices that bolster sustainability performance, further advancing the field of sustainable interior design in Jordan and beyond.

**Keywords:** sustainability; environmental sustainability; economic sustainability; social sustainability; sustainable interior design practices; Jordan

## 1. Introduction

The field of sustainable interior design has witnessed a notable surge in global attention, driven by mounting environmental concerns and a heightened demand for design solutions that are both ecologically conscious and socially responsible [1,2]. Within the context of Jordan, a nation experiencing rapid urbanization and robust economic development, the integration of sustainable principles into interior design emerges as a critical imperative. This integration is crucial not only for minimizing the environmental footprint of construction and renovation projects, but also for enhancing the wellbeing of occupants within these spaces [3]. In this context, interior designers play a pivotal role as stewards of the built environment, wielding the potential to instigate positive change through the deliberate adoption of sustainable design practices.

Sustainable interior design, in its essence, encompasses a multifaceted approach that spans energy efficiency, the judicious selection of eco-friendly materials, the minimization of waste, and the conscientious consideration of social and cultural factors [4]. As the vanguards of spatial aesthetics and functionality, interior designers' attitudes and inclinations toward sustainability exert profound influence over the extent to which these practices permeate their projects [5]. Indeed, understanding the perspectives, motivations, and challenges that interior designers encounter in their pursuit of sustainability is pivotal for advancing sustainable interior design in Jordan.

Sustainable interior design refers to the conscious consideration of environmental, social, and economic aspects when making design decisions. Sustainable interior design encompasses a diverse array of strategies and practices aimed at minimizing resource depletion, reducing environmental impacts, and enhancing human health and wellbeing. For instance, the integration of energy-efficient design elements, such as LED lighting systems and passive cooling techniques, can substantially reduce a building's energy consumption and environmental footprint. A notable example can be found in the study by Kent et al. [6], where the implementation of indoor fans in a zero-energy office building in Singapore resulted in a 32% reduction in cooling energy consumption while maintaining high levels of human thermal comfort. These energy-efficient solutions not only mitigate environmental impact but also contribute to cost savings and occupant satisfaction.

Furthermore, sustainable interior design principles emphasize the selection of eco-friendly materials with reduced environmental footprints, such as recycled and reclaimed materials. For instance, the use of reclaimed wood flooring not only conserves virgin timber resources, but also imparts a unique character to interior spaces. Waste reduction strategies, including recycling and responsible sourcing, play a pivotal role in minimizing construction and demolition waste, thereby lessening the burden on landfills.

Creating healthier indoor environments is another paramount facet of sustainable interior design. This involves incorporating strategies to improve indoor air quality, such as the use of low-VOC (Volatile Organic Compound) paints and finishes, as well as the integration of natural ventilation systems. The result is an interior environment that promotes occupant wellbeing, productivity, and overall quality of life [7,8]. The principles of sustainable interior design encompass energy efficiency, use of eco-friendly materials, waste reduction, and the creation of healthier indoor environments [9,10]. Sustainable interior design is a holistic approach that integrates environmentally responsible practices, social considerations, and economic viability into the design and construction of interior spaces. Environmental interior design, guided by key principles that drive its sustainability, aims to create healthy, resource-efficient, and environmentally friendly environments that promote occupant wellbeing while minimizing negative impacts on the planet. This approach prioritizes the use of eco-friendly materials, energy-efficient systems, and sustainable construction practices to minimize negative impacts on the environment, emphasizing environmental responsibility. Simultaneously, it focuses on promoting social wellbeing by creating spaces that enhance occupant health, comfort, productivity, and inclusivity, while ensuring accessibility for all. Lastly, economic viability is a crucial principle, aiming to optimize resource utilization, reduce operational costs, and provide long-term economic benefits, making sustainable interior design both environmentally conscious and economically advantageous [11].

Sustainable interior design offers a range of benefits. Firstly, it provides environmental advantages by utilizing sustainable materials, minimizing energy consumption, and reducing waste generation, thereby contributing to lower carbon footprints and the preservation of natural resources. Secondly, it promotes occupant health and wellbeing through the incorporation of elements, such as good indoor air quality, natural lighting, ergonomic design, and biophilic elements, resulting in improved health, productivity, and satisfaction. Lastly, sustainable interior design facilitates cost savings by implementing energy-efficient systems, using durable materials, and optimizing resource utilization, leading to reduced operational costs and long-term financial savings [11].

Sustainable interior design requires careful consideration of various factors. Firstly, materials play a crucial role, and designers should prioritize the selection of eco-friendly, recycled, or locally sourced materials that have a reduced environmental impact throughout their life cycle. Energy efficiency is another key consideration, involving the incorporation of energy-efficient lighting, HVAC systems, appliances, and passive design strategies to minimize energy consumption. Indoor environmental quality must be ensured, encompassing aspects, such as good air quality, acoustic comfort, thermal comfort, and proper ventilation, to support occupant health and wellbeing. Water conservation is also important, and designers should implement water saving fixtures, greywater systems, and efficient water-use strategies within interior spaces. Lastly, waste management practices are essential, and designers should promote recycling, reuse, and responsible waste management throughout the construction process and the entire life cycle of the interior space. By addressing these considerations, sustainable interior design can be achieved, resulting in environmentally responsible and healthy spaces. Sustainable interior design integrates environmental, social, and economic considerations to create interior spaces that are environmentally responsible, socially inclusive, and economically viable. By adhering to principles, such as environmental responsibility, social wellbeing, and economic viability, sustainable interior design offers numerous benefits, including reduced environmental impacts, improved occupant health and wellbeing, and long-term cost savings. By considering key factors, such as materials, energy efficiency, indoor environmental quality, water conservation, and waste management, interior designers can contribute to a more sustainable and resilient built environment.

## 2. Literature Review

### 2.1. Attitudes and Sustainable Design Adoption

Kineber et al. [12] conducted a study that explored sustainable interior design implementation barriers in Egypt. The study aimed to identify and analyze obstacles hindering the implementation of sustainable interior design. The findings revealed that governmental obstacles were the most pressing, followed by issues related to information, knowledge, awareness, technology, training, attitudes, the market, and economics. The study emphasized the need for enhanced training of interior architects and designers, as well as increased awareness among stakeholders through government support and regulation. Another important study that explored the significance of material selection and environmental standards in achieving sustainable interior design was conducted by Yan [13]. It emphasized the importance of incorporating material and technical skills, architectural aesthetic principles, and meeting the functional and comfort needs of occupants. The study suggested that interior design should not only fulfill functional requirements, but also reflect historical context, architectural style, environmental atmosphere, and spiritual factors. It recommended the adoption of green ecological design concepts and the application of new materials and technologies to improve the ecological construction level of interior environments.

A study was conducted by Ashour et al. [14] on a comprehensive review of deterrents to the practice of sustainable interior architecture and design in Egypt. This study focused on the concept of Sustainable Interior Architecture and Design (SIAD) and its significance in achieving sustainable development goals. It identified and categorized 61 deterrents to SIAD practice, including economic, attitude, knowledge, awareness, market, information, technology, education and training, and government and professional bodies. The study highlighted the need for further research to address these identified deterrents.

Several factors influence interior designer's attitudes toward sustainable interior design. One of the key factors is education and training. Designers with a strong background in sustainability tend to hold more positive attitudes and are more likely to prioritize sustainable design features [15]. Client preferences, project budget constraints, and availability of sustainable materials also play a significant role in shaping designers' attitudes and decision making [16,17]. Şule et al. [18] conducted a cross-cultural study that explored the

integration of green building design principles into interior architecture education. It utilized a global teamwork project, incorporating a green building assessment system (LEED checklist) and examples of vernacular architecture as precedents. The study found a gap in students' understanding of green building between developing and developed countries and suggested that collaborative project experiences could bridge this gap and facilitate the exchange of technical and cultural information related to sustainability. Another study that tackled education as a key factor that has a sustainability impact on interior design practices was conducted by Rashdan [19]. This study focuses on the role of designers in deciding sustainable solutions and incorporating sustainable principles in interior design. It identifies barriers to the implementation of sustainable design, including cost considerations, limited material and system selection, lack of experience, and the absence of circular economy principles. The circular economy framework, which emphasizes "reduce, reuse, and recycle" (the 3Rs), plays a pivotal role in sustainable interior design. By adopting circular economy principles, designers can contribute to the reduction in waste, the efficient use of resources, and the creation of environmentally responsible and economically viable interior spaces. The study emphasizes the importance of sustainability and recommends that environmentally responsible interior designers, with the support of associations and design firms, collaborate to develop specific standards covering different aspects of sustainable interior design.

Attitudes are fundamental determinants of behavior, and they significantly influence the adoption of sustainable design practices. Positive attitudes toward sustainability are linked to a higher likelihood of integrating sustainable strategies into design projects. Understanding the factors that shape interior designers' attitudes is essential for encouraging the widespread adoption of sustainable interior design practices [20]. Bacon [21] examined the attitudes of interior designers toward sustainable interior design practices and the barriers they encounter. She evaluated the perceived barriers in three areas: project capabilities, transition to sustainability, and knowledge and skills associated with sustainable design. The study concludes that attitudes toward sustainable interior design practices are positive, and factors affecting project capabilities are identified as the biggest obstacle. The findings suggest a correlation between attitudes and perceived barriers, indicating that a positive attitude contributes to overcoming barriers and vice versa. Another study tackled the issue of attitude conducted by Máté [22] who investigated the attitudes and decision-making processes of interior designers in Australia regarding sustainable design. It reveals a contradiction in interior design practices where designers often do not align their behavior and actions with their professed attitudes toward sustainability. The study highlights the reliance of many designers on clients or external agencies to prioritize sustainable design approaches. It also identifies a lack of confidence in the designers' own knowledge and the information provided by suppliers regarding sustainable issues. The findings emphasize the need for additional resources, information, and education to support interior designers in making choices consistent with sustainable design principles.

### 2.2. Sustainable Interior Design Practices in Jordan

The Jordanian interior design sector faces unique challenges and opportunities in adopting sustainability principles. With a growing emphasis on sustainable development, there is an increasing awareness of the need for environmentally and socially responsible interior design practices in the country. Government initiatives, such as the National Green Building Strategy, provide a supportive framework for integrating sustainability principles. A study conducted by Hussein [23] discussed the challenges and opportunities faced by the Jordanian interior design sector in adopting sustainable practices. It highlights the growing emphasis on sustainable development and the need for environmentally and socially responsible interior design. The study emphasizes the role of government initiatives, such as the National Green Building Strategy, in providing a supportive framework for integrating sustainability principles into interior design practices in Jordan.

Altamimi et al. [24] explored the experiences of interior design professionals in Jordan regarding social sustainability in workplace design. Her study examined the four dimensions of physiological health and comfort, efficiency and ergonomics, privacy and social interaction, and spatial organization. The study provides valuable insights for the development of workplace design guidelines that prioritize social sustainability parameters within the built environment. Obeidat et al. [25] conducted a study focusing on the role of sustainable interior design and its impact on customer behavior in commercial environments. The study examined the interior environment quality in terms of sustainable design and its effects on building performance, efficiency, and services. It highlighted the importance of integrating sustainable design into interior environments to enhance customer behavior and improve the quality, efficiency, and effectiveness of commercial buildings. The study addressed the positive role of sustainable interior design in influencing customer behavior and raising the performance and services of commercial environments.

Matarneh [26] conducted an exploratory study that investigated sustainability in traditional and vernacular Jordanian architecture. Based on the analysis of three key global sustainable building certification systems: LEED, BREEAM, and Green Globes, the study provided a comprehensive framework that can serve as an instrumental tool to design and assess traditional and vernacular buildings in Jordan. The framework considered the unique characteristics of Jordan's architectural heritage, climate, cultural values, and socio-economic context. It incorporated principles of sustainability, preservation, and functionality to ensure the compatibility of traditional buildings with modern requirements. The framework encompassed various aspects, including architectural design, structural integrity, energy efficiency, water conservation, indoor environmental quality, and cultural significance. By applying this framework, architects, interior designers, engineers, and heritage conservation experts can effectively evaluate the condition of existing buildings, identify areas for improvement, and develop strategies for their sustainable adaptation and preservation. This framework not only supported the conservation of Jordan's architectural heritage, but also contributed to the creation of environmentally responsible, culturally sensitive, and economically viable built environments in the country.

Altomonte et al. [27] conducted a study in the UK to assess occupant satisfaction in BREEAM-certified and non-certified office buildings. They analyzed data from questionnaires and found that BREEAM certification, in itself, did not significantly influence building and workspace satisfaction. The study highlights the need to delve deeper into the factors affecting occupant satisfaction in certified buildings.

Altomonte et al. [28] extended this line of research by investigating the relationship between IEQ credits achieved in LEED-rated office buildings and occupant satisfaction. Their study, based on a substantial dataset, concluded that achieving specific IEQ credits did not significantly increase satisfaction with corresponding IEQ factors. Additionally, the rating level and the product and version of certification did not substantially affect workplace satisfaction. This research suggests that while green certifications aim to improve IEQ, several factors may contribute to variations in occupant satisfaction [28].

Licina and Yildirim [29] explored the impact of WELL certification on occupant satisfaction, productivity, and health. They compared occupants' satisfaction with IEQ in non-WELL (BREEAM and conventional) and WELL-certified office buildings. Their findings revealed that transitioning to WELL buildings led to a statistically significant increase in building and workspace satisfaction in two out of three building pairs. Notably, WELL certification positively affected parameters such as building cleanliness and furniture. However, there was no significant difference in satisfaction with noise and visual comfort. Additionally, the level of certification did not consistently correlate with overall building satisfaction scores. The study also examined Sick Building Syndrome (SBS) symptoms and self-reported productivity, revealing that, except for the symptom of tiredness, there were generally insignificant differences between WELL and non-WELL buildings. The impact of COVID-19 measures was noted to influence the self-reported work abilities of occupants [29].

These studies shed light on the intricate relationship between sustainable building certifications and occupant satisfaction with IEQ. While these certifications aim to improve various aspects of the indoor environment, including air quality, lighting, and comfort, the degree to which they impact occupant satisfaction can vary. Factors, such as the specific certification system, building design, and occupant expectations, may all play a role in determining occupant satisfaction levels. Further research is needed to explore these dynamics and optimize sustainable building practices to enhance both environmental performance and occupant wellbeing. The reviewed studies shed light on the current state of sustainable interior design practices in Jordan. They highlight the challenges and opportunities faced by the Jordanian interior design sector in adopting sustainability principles and emphasize the importance of government initiatives in supporting sustainable design. These challenges include the limited availability of sustainable materials and products, higher initial costs, lack of awareness, and resistance from clients who prioritize cost efficiency over sustainability [30,31]. These findings contribute to a better understanding of the status quo and the potential for further development and improvement in sustainable interior design practices in Jordan.

### 2.3. Sustainability Indicators in Interior Design

2.3.1. Environmental Sustainability Indicators

Sustainable interior design strategies play a crucial role in promoting environmentally responsible practices within the field of interior design. These strategies aim to minimize the negative environmental impacts of interior spaces while creating healthier and more resource-efficient environments. One of the key aspects of sustainable interior design is the careful selection of materials. Choosing environmentally friendly and low-impact materials, such as those that are recycled, renewable, or locally sourced, can significantly reduce the carbon footprint associated with interior design projects [32]. Additionally, considering the life cycle assessment of materials helps in understanding their environmental impact throughout their entire lifespan. Mohsen and Akash [33] found that only 5.7% of dwellings in Jordan's urban areas have been provided with wall insulation and none with roof thermal insulation. It was shown that energy savings of up to 76.8% can be achieved when polystyrene is used for both wall and roof insulation. Integrating energy-efficient measures into interior design is crucial for reducing energy consumption and minimizing the environmental footprint of buildings. Strategies, such as optimizing natural lighting, implementing energy-efficient lighting fixtures, utilizing energy-efficient appliances, and incorporating efficient HVAC systems, contribute to significant energy savings [34]. Jaber et al. [35] evaluated space heating systems used in Jordan, the benefits and costs of each system were considered, and the overall benefit-to-cost ratios were determined. Their analyses showed that heating systems based on renewable energy, are most favorable.

Prioritizing indoor environmental quality is essential for promoting occupant health and wellbeing. Sustainable interior design strategies focus on enhancing indoor air quality, acoustics, and thermal comfort. This can be achieved through proper ventilation systems, the use of low-emission materials, and the incorporation of sound-absorbing surfaces [9]. Water conservation is another critical aspect of sustainable interior design. Implementing water-efficient fixtures, such as low-flow faucets and dual-flush toilets, helps reduce water consumption within interior spaces. Additionally, incorporating water saving strategies for landscaping and utilizing greywater systems for non-potable water uses contribute to overall water conservation efforts [36]. Promoting adaptive reuse and upcycling of existing materials and furniture items reduces waste and extends the lifespan of resources. Integrating salvaged materials, repurposing furniture, and incorporating reclaimed wood or recycled materials in interior design projects contribute to sustainable practices [37]. In his article on Jordan's water strategies, Mohsen [38] recommended that water conservation in Jordan should be pursued through increased water recycling and other possible options.

Sustainable interior design strategies encompass a range of practices aimed at minimizing environmental impacts, optimizing resource efficiency, and promoting occupant

wellbeing. By incorporating strategies, such as materials selection, energy efficiency, indoor environmental quality, water conservation, and adaptive reuse, interior designers can contribute to the creation of sustainable and resilient built environments. Implementing these strategies not only benefits the environment, but also enhances occupant comfort, health, and productivity, making sustainable interior design a crucial aspect of contemporary design practice.

### 2.3.2. Economic Sustainability Indicators

Economic sustainability is a crucial aspect of interior design that focuses on creating spaces that are financially viable, cost effective, and contribute to long-term economic benefits. This section explores key economic sustainability indicators in interior design, highlighting their significance in promoting efficient resource allocation, cost savings, and economic viability. Life cycle cost analysis involves evaluating the total cost of a project over its entire lifespan, including initial costs, operating expenses, maintenance costs, and potential end-of-life costs. By conducting a thorough analysis, interior designers can make informed decisions regarding material selection, energy-efficient systems, and maintenance strategies to optimize cost effectiveness over time [39]. Incorporating energy-efficient design strategies in interior spaces can significantly impact operational costs. By integrating energy-efficient lighting, HVAC systems, and appliances, interior designers can help reduce energy consumption and lower long-term operational expenses. Choosing sustainable materials and considering their financial implications is essential for economic sustainability. Opting for durable, low-maintenance materials and exploring local sourcing options can reduce material costs, transportation expenses, and waste generation [40]. Emphasizing adaptive reuse and renovation instead of new construction can have economic benefits. Reimagining existing spaces and repurposing materials can significantly reduce construction costs, resource consumption, and waste generation [41]. Considering the return on investment is crucial for economic sustainability in interior design. This involves assessing the financial benefits and potential cost savings associated with design decisions, such as implementing energy-efficient systems, optimizing space utilization, and enhancing occupant productivity [42].

Economic sustainability indicators in interior design emphasize efficient resource allocation, cost effectiveness, and long-term economic viability. By conducting life cycle cost analyses, prioritizing energy efficiency, selecting sustainable materials, promoting adaptive reuse, and considering return on investment, interior designers can contribute to economic sustainability. These indicators not only help minimize operational expenses and construction costs, but also enhance the overall financial performance and economic value of interior design projects.

### 2.3.3. Social Sustainability Indicators

Social sustainability is an essential aspect of interior design, focusing on creating spaces that enhance the wellbeing, health, and satisfaction of occupants. This section explores key social sustainability indicators in interior design, highlighting their significance in promoting inclusive, accessible, and supportive environments. Universal design principles aim to create spaces that are accessible and usable by people of all ages, abilities, and backgrounds. In interior design, this involves incorporating features, such as barrier-free access, adaptable furniture, and inclusive layouts that accommodate diverse user needs. Prioritizing the health and wellbeing of occupants is a crucial social sustainability indicator. Interior design can contribute to improving indoor air quality by using low-emission materials, integrating proper ventilation systems, and maximizing access to natural light. Additionally, incorporating biophilic design elements and creating spaces that promote physical activity and mental wellbeing can enhance the overall health of occupants [4]. Social sustainability in interior design emphasizes creating comfortable and ergonomically sound spaces. This involves considering factors, such as ergonomic furniture design,

appropriate lighting levels, acoustic control measures, and temperature regulation, to enhance occupant comfort and productivity [43].

Interior design can foster community engagement and collaboration by creating spaces that encourage social interaction, communication, and connectivity. Design strategies, such as incorporating communal areas, flexible workspaces, and shared amenities, promote a sense of community and support social interactions among occupants [44]. Interior design should reflect and respect the cultural diversity of occupants. Incorporating elements that celebrate cultural heritage, traditions, and values contributes to a sense of belonging and inclusivity. Designing spaces that accommodate diverse cultural practices, preferences, and needs fosters a welcoming and inclusive environment [45].

Social sustainability indicators in interior design focus on creating inclusive, accessible, and supportive spaces that enhance the wellbeing and satisfaction of occupants. By incorporating universal design principles, promoting health and wellbeing, prioritizing comfort and ergonomics, fostering community engagement and collaboration, and embracing cultural sensitivity, interior designers can contribute to social sustainability. These indicators not only enhance the quality of interior spaces, but also promote a sense of belonging, connectivity, and overall satisfaction among occupants.

## 3. Methodology

This study aims to examine the current attitudes of interior designers in Jordan towards sustainable interior design practices. The objectives include understanding the potential benefits, challenges, and opportunities associated with implementing sustainability principles in interior design. To achieve these objectives, a comprehensive review of relevant academic literature and existing studies on interior designers' attitudes toward sustainable interior design practices in Jordan was developed. This literature review served as the foundation for understanding the practices, drivers, potential links to sustainability performance, and identifying research gaps and areas of focus. Based on this literature, a research instrument was developed specifically for the Jordanian context.

The classification of interior design practices according to these sustainability dimensions serves a dual purpose: elucidating and expanding our comprehension of the adoption of sustainability within the current landscape of interior design practices. It underscores the interplay between theory and practice, as theoretical frameworks bridge the gap between conceptual understanding and practical application, thereby providing actionable insights and directions that can be harnessed by interior designers.

### 3.1. Survey Design and Methodology

### 3.1.1. Survey Design

The survey conducted in this study was meticulously designed to gather data on the attitudes of interior designers in Jordan toward sustainable interior design practices. The survey aimed to capture a comprehensive understanding of respondents' perspectives on various aspects of sustainability in interior design. This section provides a detailed overview of the survey design, including the Likert scale used, the number of points on the scale, the questions asked, and their format.

### 3.1.2. Likert Scale

A 5-point Likert scale was employed in the survey. This scale ranged from 1 (strongly disagree) to 5 (strongly agree), with a midpoint of 3 (neutral). Respondents were asked to indicate their level of agreement with statements related to sustainable interior design practices using this scale. The Likert scale allowed for nuanced responses, enabling respondents to express their opinions and perceptions accurately.

### 3.1.3. Survey Questions

The survey was structured into three main parts, each targeting a specific aspect of sustainable interior design. Below is a brief overview of the key questions and topics covered in each part:

Part 1: Respondent Background.

This section gathered information about the respondents' backgrounds, including their education, professional affiliations or memberships, experiences, and roles in the interior design field. It helped in establishing a demographic profile of the participants.

Part 2: Awareness and Understanding of Sustainability.

This part explored the respondents' level of awareness and understanding of the concept of sustainable interior design. Questions in this section assessed the extent to which respondents were familiar with sustainability principles and their potential implications.

Part 3: Implementation of Sustainability Principles.

The third part was divided into three sections, each addressing a specific dimension of sustainability: environmental, economic, and social. Within each section, respondents were presented with statements related to sustainability indicators in interior design practices. They were asked to use the Likert scale to indicate their level of agreement or disagreement with these statements, assessing the degree to which sustainability principles were integrated into their practices.

### 3.1.4. Format

The survey was administered electronically, allowing respondents to complete it at their convenience. It was structured using an online platform, making it easily accessible to participants. The questions were presented in a clear and straightforward format to ensure that respondents could provide accurate and meaningful responses.

The careful design of the survey, including the choice of a 5-point Likert scale, specific questions, and an online format, aimed to collect comprehensive and valuable data regarding interior designers' attitudes toward sustainable interior design practices in Jordan. These data were crucial in achieving the objectives of the study and contributing to the understanding of sustainability in the field of interior design. The following research questions have been identified for this study:

i.     To what extent are interior designers in Jordan aware of the concept and understand the principles and objectives of sustainable interior design?
ii.    To what extent do interior designers in Jordan implement sustainability principles in their design practices?
iii.   How frequently do interior designers in Jordan incorporate environmentally sustainable elements, such as energy-efficient lighting, recycled materials, or indoor air quality management, in their designs?
iv.    To what extent do interior designers consider the economic aspects of sustainability, such as cost effectiveness and long-term savings, when making design decisions?
v.     How do interior designers in Jordan address social sustainability factors, such as user wellbeing, accessibility, and community engagement, in their projects?

### 3.2. Population and Sample

The participants in this study consisted of experts and professionals employed in Jordanian interior design companies and architecture/engineering firms that offer interior design services. A purposive sampling approach was used to select a sample of specialists (interior designers and architects, engineers, sustainability consultants, regulators, contractors and developers, and suppliers) who were mostly members of professional associations (JEA, JIDA, and JCCA). The sample size is relatively small with 118 respondents.

The collection of primary data was conducted through a questionnaire. The questionnaire was structured into three parts. The first part investigates the respondents' backgrounds including their education, professional affiliations or memberships, experiences, and roles. The second part is to investigate the respondents' level of awareness and

understanding of the concept of sustainable interior design and the level of implementation of sustainability principles into interior design. The last part consisted of three sections to investigate the level of implementation of sustainability principles into interior design practices, namely, the environmental, economic, and social sustainable dimensions. Likert scale was used in the second and the third parts of the questionnaire, and the respondents were asked to indicate how much they agreed or disagreed with statements about their implementation of sustainability indicators in interior design practices. The final sample included 118 experts and professionals employed in Jordanian interior design companies. This part of the study delved into the degree to which sustainability principles have been integrated into interior design practices, encompassing a comprehensive assessment of 28 sustainability factors categorized into three dimensions: twelve environmental interior design factors, eight social factors, and eight economic factors. Figure 1 depicts the research process that the authors will follow.

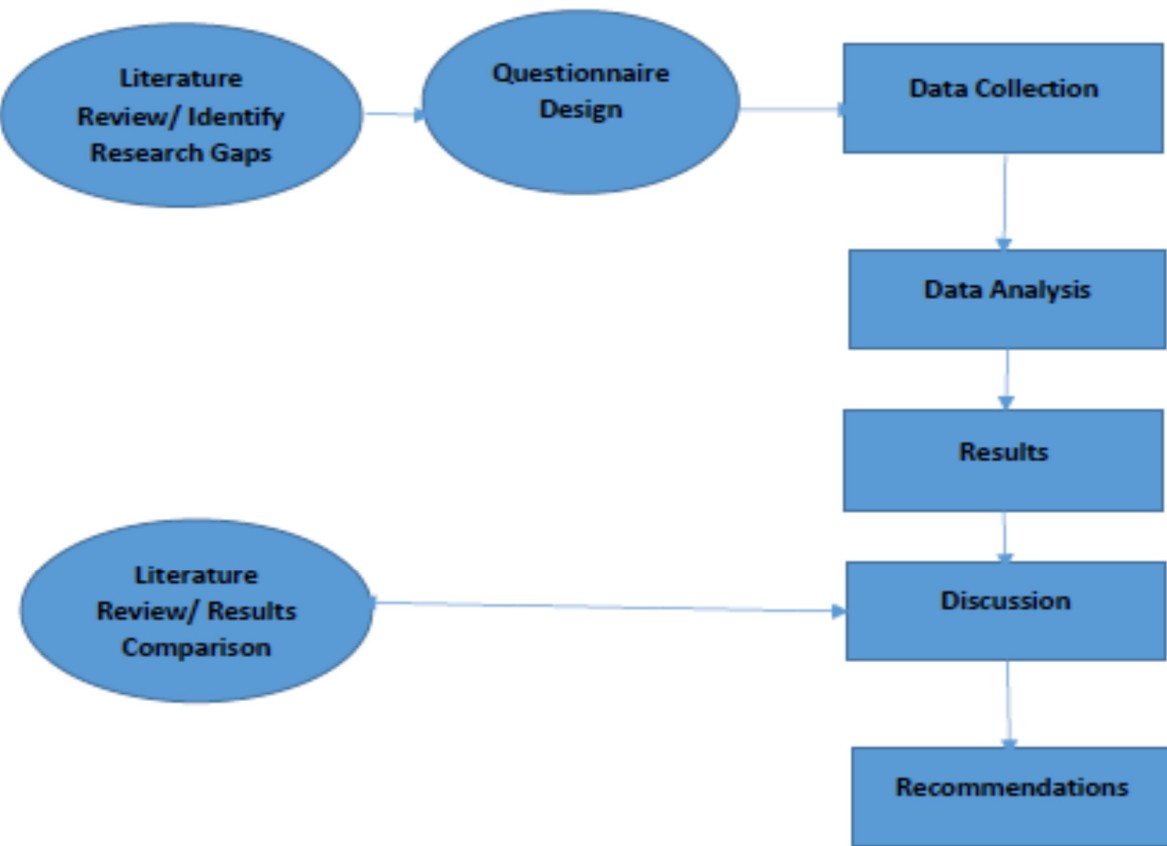

**Figure 1.** Research process.

*3.3. Sample Profile*

The demographic profile of the respondents was analyzed, as shown in Figure 2, which illustrates the educational background of the respondents, with the majority, 72.9%, holding a Bachelor's degree. Additionally, 6% of the respondents had obtained a Master's or Ph.D. degree.

Regarding professional affiliations, 66.3% of the respondents were members of professional organizations. Specifically, 31% were members of the Jordanian Interior Designers Association (JIDA), 26.3% were members of the Jordanian Engineers Association (JEA), and 9% were members of the Jordanian Construction Contractors Association (JCCA).

In terms of professional experience, 44.1% of the respondents had over 15 years of experience, while 36.4% had between 10 and 15 years of experience. Furthermore, 15.2% had between 5 and 9 years of experience, and only 4.3% had 5 years or less of experience.

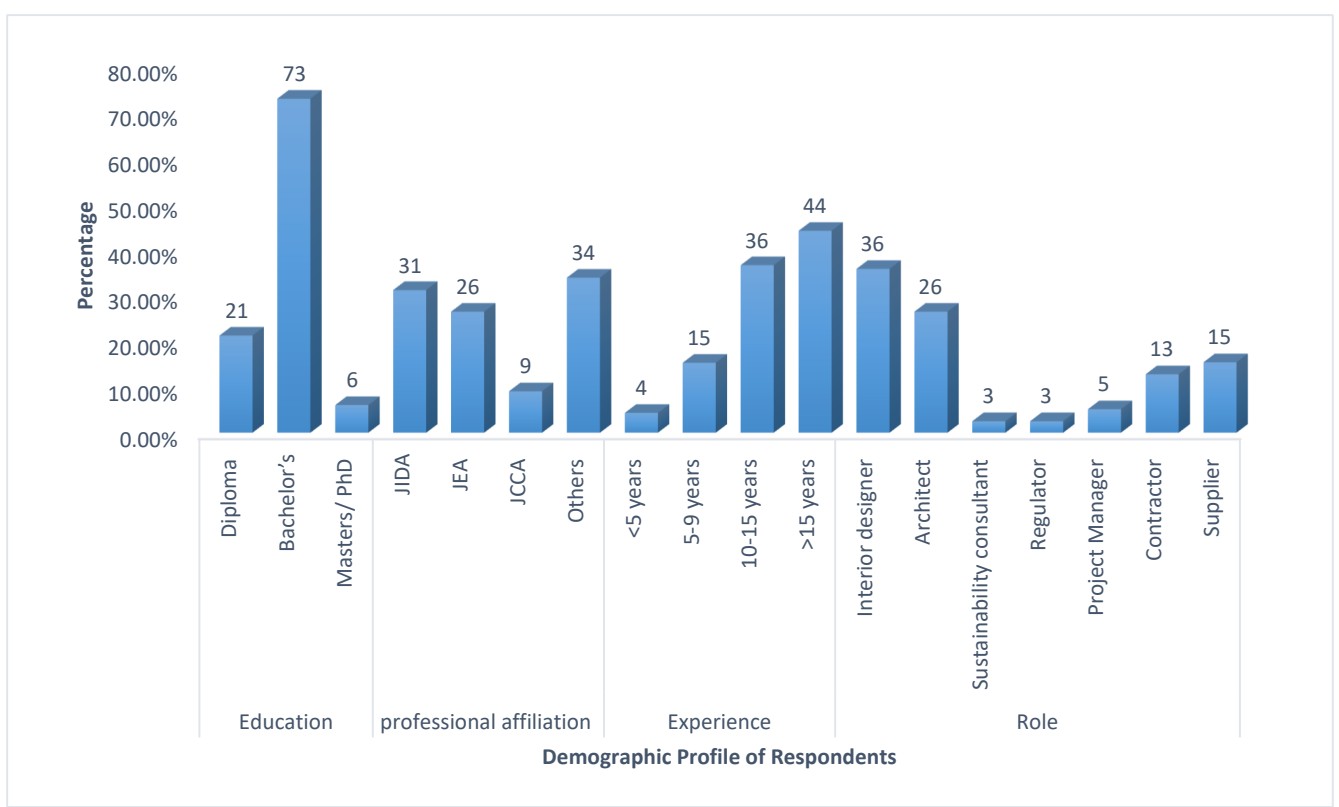

**Figure 2.** Demographic profile of the respondents.

In relation to professional roles, 35.6% of the respondents identified themselves as interior designers, while 26.3% were architects. Other roles included are sustainability consultants, regulators, project managers, contractors, and suppliers that account for 2.5%, 2.5%, 5.1%, 12.7%, and 15.3%, respectively.

## 4. Results

### 4.1. Level of Professionals' Awareness about the Concept of Sustainable Interior Design

As shown in Figure 3, the results of the investigation into professionals' awareness of the concept of sustainable interior design indicate a notable level of awareness among the respondents. When asked about the importance of sustainable interior design practices to reduce negative impacts on the environment and to preserve natural resources, the responses were highly positive. Specifically, 50% of the respondents expressed a very high level of awareness, 34.8% indicated a high level of awareness, 8.7% had an average level of awareness, and only 6.5% had a low level of awareness. These findings demonstrate a significant recognition among professionals regarding the significance of sustainable interior design practices in conserving the natural environment and resources.

The responses regarding participants' perception of sustainable interior design practices as more than just a passing trend aligns with their level of awareness, displaying a highly positive sentiment. Herein, 42.4% of respondents strongly agreed with the statement, indicating a firm conviction in the enduring nature of sustainable interior design practices. Additionally, 39% of respondents expressed a high level of agreement, further reinforcing the notion that these practices are here to stay. A smaller proportion, 8.7%, indicated an average level of agreement, while only 6.5% agreed with a low level of conviction. These findings underscore the widespread belief among participants that sustainable interior design practices hold long-term significance and are not merely transient trends. The results of the participants' responses regarding their agreement with the statement "Sustainable interior design benefits the health and welfare of building occupants" indicate a strong consensus among the participants.

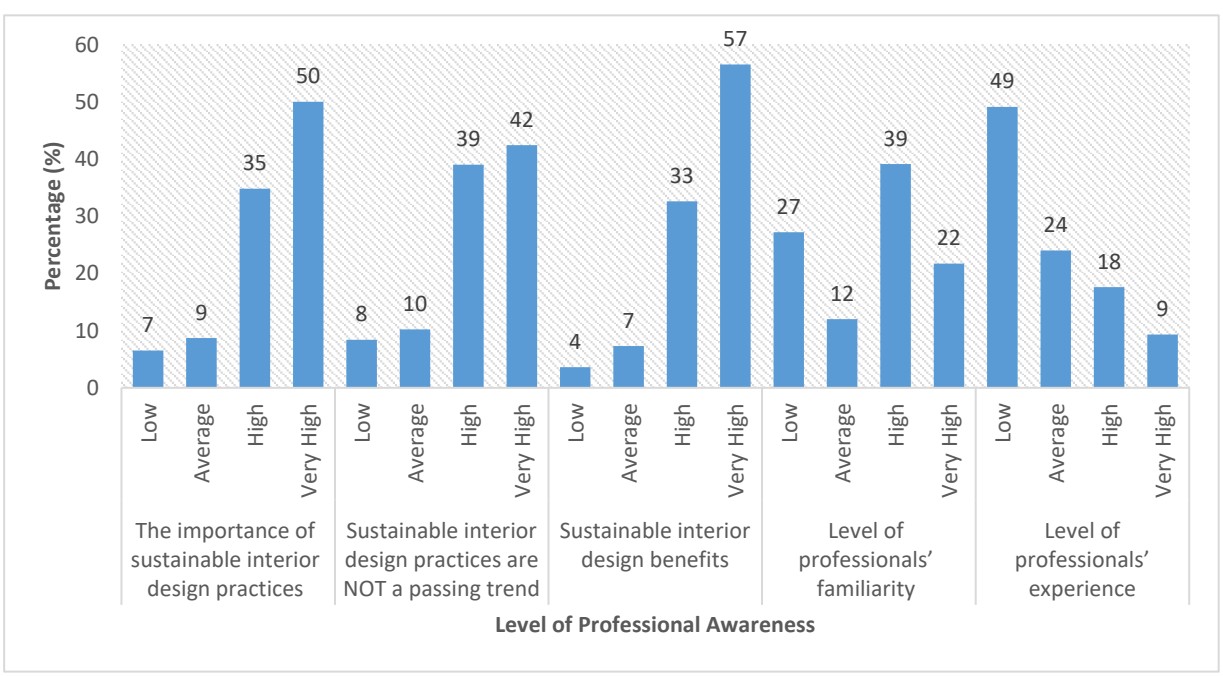

**Figure 3.** Professionals' awareness of the concept of sustainable interior design.

Figure 3 shows that a significant majority, 56.5%, expressed a very high level of agreement, indicating a strong belief in the positive impact of sustainable interior design on the health and welfare of building occupants. An additional 32.6% of participants responded with a high level of agreement, further supporting the notion that sustainable interior design practices have beneficial effects on occupants' wellbeing.

A smaller proportion, 7.3%, indicated an average level of agreement, suggesting some uncertainty or mixed opinions on the matter. Only 3.6% of participants expressed a low level of agreement, indicating a minority view that sustainable interior design may not significantly contribute to the health and welfare of building occupants.

Overall, these results highlight a widespread recognition among the participants that sustainable interior design practices play a crucial role in enhancing the health and wellbeing of individuals residing or working within buildings.

The analysis of professionals' familiarity with the concept of sustainable interior design, as presented in Figure 3, reveals a noteworthy level of familiarity among the respondents. When queried about their knowledge of sustainable interior design, the responses generally indicated a positive understanding. Specifically, 21.7% of the participants expressed a very high level of familiarity, while 39.1% indicated a high level of familiarity. Additionally, 12% reported an average level of familiarity and 27.3% stated a low level of familiarity.

These findings demonstrate that professionals possess a significant awareness of the importance of sustainable interior design practices. However, it is worth noting that while many respondents acknowledged the significance and necessity of incorporating sustainability principles into interior design projects, there appears to be a lack of familiarity with effectively applying sustainability concepts and strategies in their designs.

This suggests that although professionals recognize the importance of sustainability, there may be a gap in their practical understanding and implementation of sustainable principles within their design practices. Further efforts may be required to bridge this gap and enhance professionals' ability to effectively integrate sustainability into their interior design projects.

The results of the analysis pertaining to professionals' experience with sustainable interior design practices, as depicted in Figure 2, indicate that the majority of participants possess a relatively low level of experience in this area.

Specifically, only 9.3% of the professionals reported having a very high level of experience with sustainable interior design practices. This suggests that a small proportion of participants have extensive hands-on experience and a deep understanding of implementing sustainability principles in their design work.

Furthermore, 17.6% of the professionals indicated a high level of experience, signifying a moderate level of familiarity and practical application of sustainable interior design practices. On the other hand, a higher percentage of participants, 24%, reported having an average level of experience, suggesting a more limited exposure to sustainable design concepts and strategies.

The majority of professionals, comprising 49.1% of the respondents, stated a low level of experience with sustainable interior design practices. This indicates a significant gap in their practical exposure and understanding of implementing sustainability in their design projects.

These findings highlight the need for further education, training, and professional development opportunities to enhance professionals' experience and expertise in sustainable interior design practices. Closing this experience gap is essential for promoting more widespread adoption of sustainable principles within the field of interior design.

This part of the study investigated sustainable interior design practitioners' attitudes and found that professionals possess a significant awareness of the importance of sustainable interior design practices. However, it is worth noting that while many respondents acknowledged the significance and necessity of incorporating sustainability principles into interior design projects, there appears to be a lack of familiarity with effectively applying sustainability concepts and strategies in their designs. This suggests that there may be a gap in their practical understanding and implementation of sustainable principles within their design practices. Further efforts may be required to bridge this gap and enhance professionals' backgrounds through education and training. Designers with a strong background in sustainability tend to hold more positive attitudes and are more likely to prioritize sustainable design features in their projects.

### 4.2. Level of Implementation of Sustainability Principles into Interior Design Practices

The descriptive analysis involved calculating key statistical measures including the mean, standard deviation, and implementation level. The implementation level was determined using a 5-point Likert scale, ranging from 1 (strongly disagree) to 5 (strongly agree). Cumulative values were then categorized into three groups: low (1–2.00), moderate (2.09–3.67), and high (3.68–5). The evaluation of sustainable indicators in interior design practice was conducted using three levels: low, medium, and high. The formula used to determine the length of the interval for each level was: interval length for level = (maximum value of the scale − minimum value of the scale)/(number of levels).

### 4.2.1. Environmental Sustainability

In Figure 4, the mean scores are presented for each sustainability indicator in interior design practices, specifically focusing on the environmental dimension of sustainability.

The analyzed statistics for environmental sustainability indicators in interior design practices reveal interesting findings. The overall implementation of environmental practices was assessed to be at a medium level, with a mean score of 3.0. This suggests that there is room for improvement in incorporating sustainable practices into interior design projects.

Among the specific indicators assessed, some practices stood out with high implementation scores. "Energy efficiency" received the highest mean score of 4.25, indicating that it is widely adopted and prioritized in sustainable interior design. This is followed closely by "Water efficiency" with a mean score of 3.90 and "Indoor Air Quality" with a mean score of 3.85. These findings reflect the significance placed on energy and water conservation, as well as the importance of providing healthy indoor environments. It is important to acknowledge the significant role that suppliers of eco-friendly systems, especially in the

private sector, play. These suppliers are instrumental in delivering and installing systems that prioritize energy efficiency, water conservation, and indoor air quality.

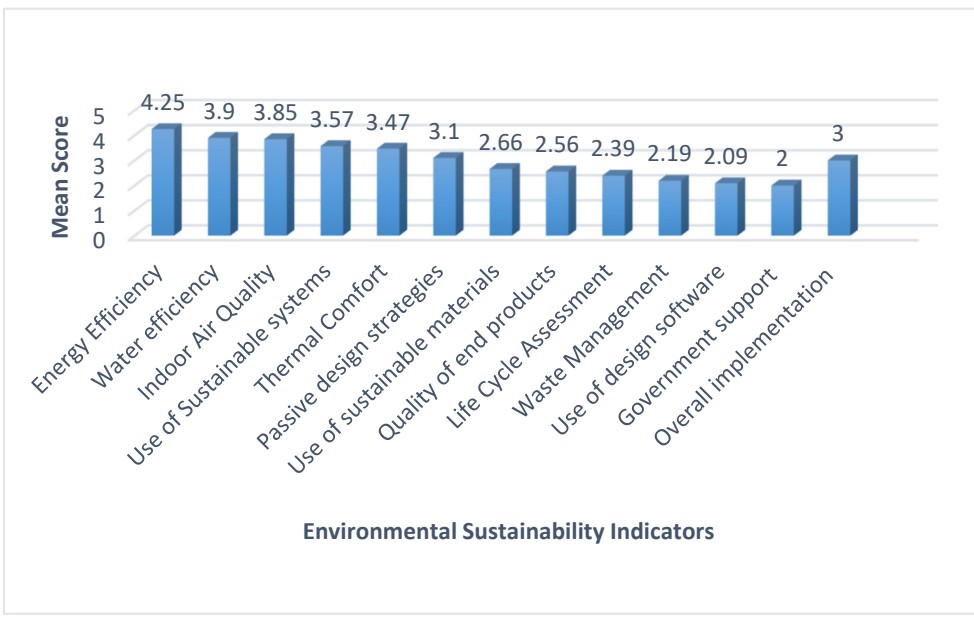

**Figure 4.** Environmental sustainability indicator in interior design practices.

On the other hand, several indicators were found to have a medium level of implementation. "Passive design strategies" such as daylighting and natural lighting obtained a mean score of 3.10, suggesting moderate adoption. The use of sustainable and environmentally friendly materials received a mean score of 2.66, indicating that there is room for improvement in incorporating these materials in interior design projects. Other indicators, such as the "Quality of end products" (mean score of 2.56), "Life Cycle Assessment" (mean score of 2.39), and "Waste Management" (mean score of 2.19), also highlight areas that could benefit from increased attention and implementation.

Furthermore, certain indicators were found to have a relatively low level of implementation. The use of "environmental design software" received a mean score of 2.09, indicating limited adoption in interior design practices. The aspect of "Government rules and regulations supporting the company's environmental practices" had the lowest mean score of 2.0, suggesting a lack of sufficient governmental support in promoting sustainable design practices.

4.2.2. Economic Sustainability

In Figure 5, the mean scores are presented for each sustainability indicator in interior design practices, specifically focusing on the economic dimension of sustainability.

The statistics for economic sustainability indicators in interior design practices reveal interesting findings. The mean scores provide insight into the level of implementation for various factors related to economic sustainability. The indicators of "Cost-effectiveness", "Life Cycle Cost Analysis", and "Value Engineering" received low mean scores of 2.05, 1.98, and 1.89, respectively. These results suggest that there is room for improvement in incorporating cost-effective strategies, conducting life cycle cost analysis, and implementing value engineering practices in interior design projects. Enhancing these aspects can contribute to maximizing economic efficiency and optimizing the use of resources.

On the other hand, the indicators of "Resource Efficiency", "Procurement Strategies", and "Operational Efficiency" obtained high mean scores of 4.27, 3.88, and 4.16, respectively. These findings demonstrate a strong level of implementation in terms of utilizing resources efficiently, adopting strategic procurement approaches, and optimizing operational pro-

cesses. By prioritizing resource efficiency and operational effectiveness, interior design practices can reduce waste, minimize costs, and enhance overall economic sustainability.

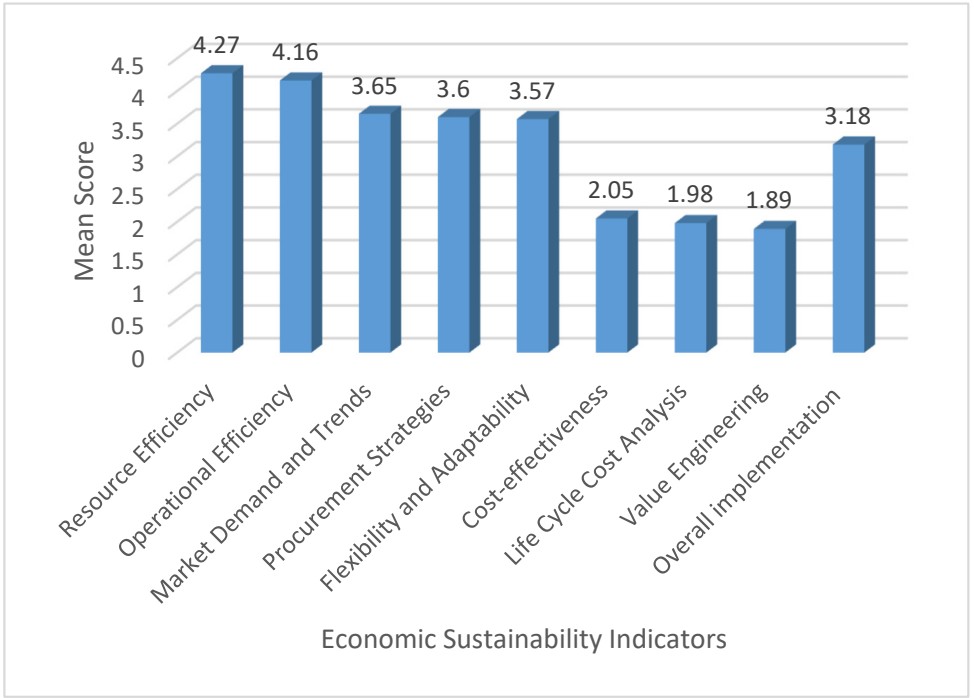

**Figure 5.** Economic sustainability indicator in interior design practices.

The indicators of "Flexibility and Adaptability" and "Market Demand and Trends" received mean scores of 3.57 and 3.65, respectively, indicating a medium level of implementation. This implies that there is a moderate focus on incorporating flexible and adaptable designs that can accommodate changing needs and market demands. Additionally, considering market trends ensures that interior design practices remain relevant and responsive to evolving end-users' preferences and expectations.

In conclusion, while interior design practices demonstrate a strong level of implementation in certain economic sustainability indicators, such as resource efficiency, procurement strategies, and operational efficiency, there is a need for improvement in other areas, such as cost effectiveness, life cycle cost analysis, and value engineering. By addressing these areas, interior design practices can further enhance their economic sustainability, optimize resource utilization, and achieve greater cost savings in their projects.

### 4.2.3. Social Sustainability

In Figure 6, the mean scores are presented for each sustainability indicator in interior design, specifically focusing on the social dimension of sustainability.

The analyzed statistics for social sustainability indicators in interior design practices reveal interesting findings.

The indicator "Designs align to laws and regulations by the Ministry of Labor" obtained a high mean score of 4.37. This indicates a strong level of implementation in ensuring compliance with relevant laws and regulations set by the Jordanian government. Next, with a mean score of 4.22, safety and security are given significant importance in interior design practices. This suggests that companies prioritize creating spaces that address safety concerns, such as proper lighting, clear wayfinding, and adherence to building codes and regulations. Designing for safety and security promotes occupant confidence and wellbeing.

Health and Wellbeing with a mean score of 4.10 indicates a strong emphasis on promoting health and wellbeing in interior design practices. This highlights the recognition

of the impact of the built environment on occupant health. Indoor Environmental Quality with a mean score of 3.98 demonstrates a focus on providing good indoor environmental quality in interior design practices. This includes factors, such as proper ventilation, effective acoustics, optimal thermal comfort, and adequate lighting. By prioritizing indoor environmental quality, interior designers create spaces that promote occupant comfort and productivity.

In conclusion, while the overall implementation of social sustainability indicators in interior design was found to be at a high level, the study reveals areas for improvement. Specifically, the findings highlight the need for greater emphasis on incorporating universal design principles to ensure accessibility and promote flexibility in interior design practices. Addressing these gaps can contribute to creating more socially sustainable interior spaces that accommodate diverse users and support long-term usability and adaptability.

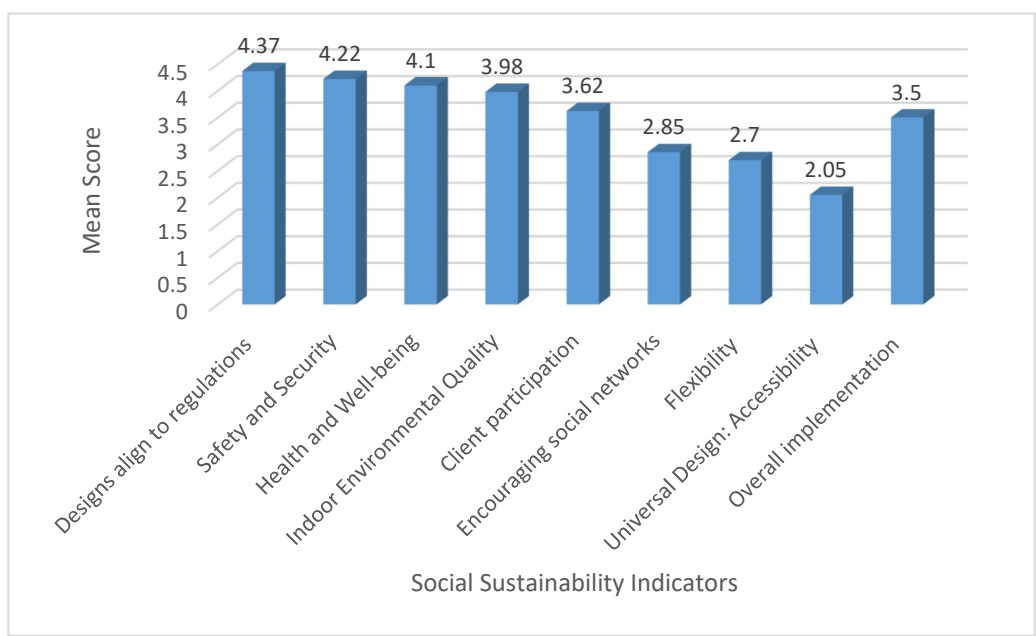

**Figure 6.** Social sustainability indicator in interior design.

## 5. Discussion

Our study provides valuable insights into the attitudes of interior designers in Jordan regarding sustainable interior design practices. Our research revealed a significant level of awareness among interior designers in Jordan regarding the importance of sustainable interior design practices. Approximately 85% of respondents demonstrated a high level of awareness, which aligns with the findings of Falcone [46] in their study on interior designers in the United States. This suggests that awareness of sustainability principles is not confined to specific geographical regions, but is a common thread in the interior design profession worldwide.

Interestingly, our study found that 81% of participants perceive sustainable interior design practices as more than just passing trends. This perception mirrors the findings of Yu et al. [47], who discovered a similar sentiment among architects regarding sustainable architecture. It is encouraging to see that professionals in the built environment increasingly view sustainability as a fundamental and enduring aspect of their practice.

Our research also highlighted the widespread recognition of the diverse benefits associated with sustainable interior design. Notably, 89% of participants acknowledged these benefits. This strong consensus echoes the findings of Zoufa et al. [48], who reported a similar high level of recognition among construction professionals regarding the benefits of sustainable construction practices. The consensus on benefits underscores the potential for a positive shift towards more sustainable design practices in Jordan.

Notably, while 61% of participants in our study demonstrated a high level of professional familiarity with sustainable interior design, only 27% claimed a high level of professional experience. This discrepancy aligns with findings in a study by Darling-Hammond et al. [49] on architects, which suggests that while professionals may have knowledge of sustainable principles, practical experience in implementing these principles may still be in the developmental stage. This indicates the need for more hands-on experience and projects that incorporate sustainability.

The mean scores for sustainability indicators provide interesting insights. Energy efficiency emerged as the highest environmental sustainability concern with a mean score of 4.25, corroborating the findings of Theodorson [50] in her research on energy-efficient building practices. Resource efficiency received the highest mean score (4.27) in economic sustainability, reinforcing the significance of efficient resource utilization as highlighted by Kang and Guerin [44]. In social sustainability, "Design aligns with laws and regulations by the Ministry of Labor" ranked highest with a mean score of 4.37, emphasizing the commitment to legal compliance and social responsibility, which aligns with the research of Sheehy and Farneti [51] in the context of social sustainability in construction.

These results underscore the importance of further efforts in promoting and implementing environmental sustainability in interior design practices. There is a need for increased focus on areas, such as sustainable materials, waste management, life cycle assessment, and the integration of environmental design software. Additionally, advocating for supportive government regulations and incentives can play a crucial role in driving the adoption of sustainable interior design practices.

The identified framework encompasses a range of strategies aimed at improving sustainability within interior design. These include the implementation of passive design strategies, which leverage natural elements to optimize energy efficiency and reduce environmental impact. Additionally, the framework emphasizes the importance of utilizing sustainable and environmentally friendly materials, such as recycled materials, renewable resources, and products with low embodied energy. The integration of environmental design software is also highlighted as a valuable practice for enhancing sustainability outcomes.

Furthermore, the study recognizes the critical role of Life Cycle Assessment (LCA) as a tool for reducing overall energy consumption and identifying opportunities for energy conservation. It underscores the significance of considering LCA in the decision-making process to promote energy-efficient design practices. Additionally, waste management practices, particularly through recycling, are highlighted as effective strategies for mitigating the detrimental impact of waste on the environment.

The findings regarding economic sustainability indicators in interior design practices reveal a mixed picture. On the one hand, indicators such as "Resource Efficiency", "Procurement Strategies", and "Operational Efficiency" received high mean scores, indicating a strong level of implementation in these areas. This demonstrates that interior design practices are effectively utilizing resources, adopting strategic procurement approaches, and optimizing operational processes. By prioritizing resource efficiency and operational effectiveness, these practices can reduce waste, minimize costs, and improve overall economic sustainability.

On the other hand, indicators such as "Cost-effectiveness", "Life Cycle Cost Analysis", and "Value Engineering" obtained low mean scores, suggesting that there is room for improvement in these areas. Incorporating cost-effective strategies, conducting life cycle cost analysis, and implementing value engineering practices are essential for maximizing economic efficiency and optimizing resource use in interior design projects. By considering the long-term costs and benefits of design decisions, conducting thorough cost analyses, and implementing value engineering principles, interior design practices can achieve greater cost savings, improved financial viability, and enhanced economic sustainability.

While interior design practices demonstrate a strong level of implementation in certain economic sustainability indicators such as resource efficiency, procurement strategies,

and operational efficiency, there is a need for improvement in other areas such as cost effectiveness, life cycle cost analysis, and value engineering. By addressing these areas, interior design practices can further enhance their economic sustainability, optimize resource utilization, and achieve greater cost savings in their projects.

Encouraging Social Networks with a mean score of 3.98 suggests a high level of implementation in encouraging social networks through design. This indicates that interior design practices recognize the importance of creating spaces that foster social interaction, collaboration, and community engagement. Next, flexibility with a mean score of 2.70 indicates that there is room for improvement in embracing flexibility in interior design practices. Universal Design: Accessibility with a mean score of 2.05 suggests a low level of implementation in incorporating universal design principles to ensure accessibility. There is room for improvement in creating spaces that are accessible to people of all abilities. Universal design promotes inclusivity and accommodates a diverse range of users. By considering accessibility as a fundamental aspect of design, interior designers can create spaces that cater to everyone.

The results of the study on social sustainability indicators in interior design indicate an overall high level of implementation. This suggests that interior designers and professionals are paying attention to social aspects and incorporating strategies that promote social sustainability in their projects. However, it is important to note that two specific factors, namely, Universal Design: Accessibility and flexibility, were found to have a very low level of implementation.

The low level of implementation of Universal Design: Accessibility indicates a gap in incorporating principles that ensure equal access and usability for all individuals, including those with disabilities. Universal Design promotes inclusive environments that accommodate diverse needs, and its limited implementation suggests a need for greater emphasis on accessibility considerations in interior design practices.

Similarly, the low level of implementation of flexibility highlights a potential lack of adaptability and responsiveness in interior spaces. Flexibility in design allows for the accommodation of changing needs, functions, and spatial requirements over time. The limited implementation of flexibility suggests a missed opportunity to create adaptable and versatile interior environments that can address evolving user needs and promote sustainable use of resources.

While the overall implementation of social sustainability indicators in interior design was found to be at a high level, the study reveals areas for improvement. Specifically, the findings highlight the need for greater emphasis on incorporating universal design principles to ensure accessibility and promote flexibility in interior design practices. Addressing these gaps can contribute to creating more socially sustainable interior spaces that accommodate diverse users and support long-term usability and adaptability.

To practically realize the innovative viewpoints uncovered in this study and leverage the high level of awareness and recognition of sustainable interior design principles among professionals in Jordan, there are practical strategies for interior designers in Jordan to further advance sustainable interior design practices, capitalize on their awareness and recognition of sustainability principles, and contribute to the broader global shift towards more environmentally responsible and socially inclusive design solutions.

To bridge the gap between familiarity and practical experience, it is essential to provide interior designers with hands-on training and workshops focused on the implementation of sustainability principles in real design projects. These training programs can offer practical insights into sustainable material selection, energy-efficient design strategies, and life cycle assessment methodologies. Interior designers should remain committed to continual education and research in the field of sustainable interior design. Staying updated on emerging trends, technologies, and best practices is crucial for driving innovation and improving sustainability outcomes.

Given the strong emphasis on resource efficiency and waste management, interior design practices can establish recycling initiatives within their organizations. This includes

recycling construction and demolition waste, reusing materials from previous projects, and sourcing sustainable, recycled, or locally available materials. Interior designers can explore the integration of environmental design software into their workflow. This software can assist in energy modeling, daylight analysis, and material life cycle assessments, enabling designers to make informed decisions that enhance environmental sustainability. Interior design professionals can collaborate with industry associations and advocacy groups to advocate for supportive government regulations and incentives that promote sustainable interior design. This can include tax incentives for green building projects or the establishment of sustainability standards and certifications.

To address the low level of implementation of universal design principles, interior designers should actively consider accessibility in their projects. This involves designing spaces that are inclusive and accessible to people of all abilities, which can be achieved through careful planning and the incorporation of universal design features. To improve flexibility in interior design, practitioners should prioritize adaptable and versatile design solutions. This can involve the use of modular furniture, flexible spatial layouts, and multi-functional spaces that can evolve to meet changing user needs over time.

Recognizing the importance of social sustainability, interior designers can engage with local communities and end users to understand their specific needs and preferences. This collaborative approach ensures that interior spaces not only meet functional requirements, but also foster a sense of community and inclusivity.

To enhance economic sustainability, interior designers should integrate life cycle cost analysis into their decision-making processes. This involves considering the long-term costs and benefits of design choices, which can lead to more cost effective and financially viable projects. The implementation of value engineering practices can help optimize resource use and reduce costs while maintaining design quality. Interior designers can proactively seek opportunities for value engineering to achieve economic sustainability goals.

In conclusion, our study underscores the positive attitudes of interior designers in Jordan towards sustainable interior design practices. The findings resonate with existing literature on sustainability in the built environment, indicating a global shift towards more sustainable and responsible design practices. However, there is a need to bridge the gap between familiarity and practical experience, which calls for hands-on training and opportunities to implement sustainable design principles effectively. Our research contributes to the growing body of knowledge in this field and provides a foundation for further exploration of sustainable design practices in diverse cultural contexts.

## 6. Conclusions

The study undertook a comprehensive examination of the attitudes of interior designers in Jordan towards sustainable interior design practices. The findings reveal a positive shift within the profession towards embracing sustainability as a fundamental approach rather than a passing trend. Notably, a substantial 85% of respondents showcased a high awareness of the significance of sustainable interior design in mitigating environmental impact and preserving natural resources. Furthermore, a significant 81% expressed a profound understanding that sustainable interior design practices extend beyond fleeting trends. The study also illuminated a strong awareness of the diverse benefits associated with sustainable interior design, garnering high recognition from 89% of participants. It is noteworthy that while 61% of participants demonstrated a high level of professional familiarity with sustainable interior design, only 27% claimed a high level of professional experience. Among the sustainability indicators, energy efficiency emerged as the most prioritized environmental concern, attaining a mean score of 4.25. Resource efficiency took precedence in economic sustainability, securing a mean score of 4.27. In the realm of social sustainability, the indicator "Design aligns with laws and regulations by the Ministry of Labor" stood out with the highest mean score of 4.37, underlining the commitment to compliance and social responsibility. Collectively, these findings underscore the shifting

paradigm towards sustainable interior design practices among professionals in Jordan, while also highlighting areas that warrant further attention and development.

This study makes a valuable contribution to the existing body of knowledge on sustainable interior design practices by introducing a comprehensive framework that enables the evaluation of interior design practices within the context of sustainability's three dimensions. The developed framework holds substantial value as a point of reference for Jordan's policymakers and interior designers. It can serve as a guiding resource for the implementation of a new strategic approach that integrates sustainability into their future development plans.

By applying this framework in a practical and holistic manner, Jordan can further advance its interior design industry towards sustainability. The framework serves as a valuable tool for guiding decision making, fostering innovation, and ultimately creating interior spaces that contribute positively to environmental preservation, economic efficiency, and social wellbeing. To practically apply the framework developed in this study, interior designers, policymakers, and relevant stakeholders in Jordan should consider the following steps:

a.  Begin by raising awareness about the framework and its importance among interior designers, architects, engineers, and other professionals involved in the built environment. Conduct workshops, seminars, and training sessions to educate practitioners about the framework's dimensions and indicators.

b.  Encourage interior designers to integrate the sustainability framework into their design practices. This involves a conscious effort to consider environmental, economic, and social sustainability dimensions during the design process.

c.  Promote collaboration among different professionals in the built environment, including interior designers, architects, engineers, sustainability consultants, and regulators. Cross-disciplinary collaboration can lead to more holistic and sustainable design solutions.

d.  Policymakers can use the framework to develop and update regulations and guidelines related to interior design practices. This may include incorporating sustainability criteria into building codes and permitting processes.

e.  Consider establishing certification programs or recognition schemes for interior design projects that successfully adhere to the sustainability framework. This can incentivize designers and clients to prioritize sustainability in their projects.

f.  Implement mechanisms for collecting data on the performance of interior design projects in relation to the sustainability framework. Regularly monitor and evaluate the impact of sustainable design practices to track progress over time.

g.  Integrate the sustainability framework into the curriculum of interior design programs and professional development courses. Ensure that future generations of interior designers are well equipped to apply sustainable principles in their work.

h.  Raise public awareness about the benefits of sustainable interior design. Educate clients and end users about the positive impact of sustainable design choices on their wellbeing, the environment, and long-term cost savings.

i.  Develop and disseminate case studies and best practices that showcase successful applications of the sustainability framework in interior design projects. Highlight the economic, environmental, and social benefits achieved.

j.  Recognize that the framework is not static and should evolve over time to reflect changing sustainability priorities and innovations in design practices. Encourage feedback from practitioners to refine and improve the framework as needed.

k.  Interior designers, along with professional associations, can advocate for sustainable interior design practices at local, national, and international levels. Networking with sustainability focused organizations can help expand knowledge and collaboration opportunities.

l.    Allocate resources for research, development, and implementation of sustainable interior design practices. This may involve dedicating funds for research projects, pilot initiatives, and educational programs.

It is important to acknowledge certain limitations in this study, primarily concerning the representativeness of the sample. The study's participants were drawn from the pool of experts and professionals working within Jordanian interior design companies and related architectural and engineering firms that provide interior design services. The use of a purposive sampling approach aimed to target a specialized group, including interior designers, architects, engineers, sustainability consultants, regulators, contractors, developers, and suppliers, many of whom were affiliated with professional associations like JEA, JIDA, and JCCA. Nevertheless, the study's sample size, which comprises 118 respondents, remains relatively modest in scope. Consequently, it is crucial to recognize that the findings may not fully capture the diversity of perspectives and experiences within the broader interior design community in Jordan, and generalizations beyond the study's specific sample should be made cautiously. Future research with larger and more diverse samples could provide a more comprehensive understanding of the attitudes and practices of interior designers across the entire spectrum of the industry in Jordan.

**Author Contributions:** Conceptualization, M.S.M. and R.M.; methodology, M.S.M. and R.M.; software, M.S.M.; validation, M.S.M. and R.M.; formal analysis, M.S.M. and R.M.; investigation, M.S.M. and R.M.; resources, M.S.M.; data curation, R.M.; writing—original draft preparation, M.S.M. and R.M.; writing—review and editing, M.S.M. and R.M.; visualization, M.S.M.; supervision, M.S.M.; project administration, M.S.M. All authors have read and agreed to the published version of the manuscript.

**Funding:** This research received no external funding.

**Institutional Review Board Statement:** Not applicable.

**Informed Consent Statement:** Not applicable.

**Data Availability Statement:** Not applicable.

**Conflicts of Interest:** The authors declare no conflict of interest.

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
