# Peer review of "Exploring the Interior Designers’ Attitudes toward Sustainable Interior Design Practices: The Case of Jordan"

_sustainability, doi:10.3390/su151914491_

Round 1
Reviewer 1 Report
The article is an interesting take on the subject.
Such analyzes are needed and can add value to science.
It should be noted that the article is written correctly and its quality meets the requirements of the journal. Its subject matter fits very well into the scope of the journal. Review of the literature on the subject, description of methodological assumptions, analysis of research results, scientific discussion and conclusions are satisfactory.
After entering the reviewer's comments, the article may be published.
Minor suggestions for improvement:
1. Separate Results and Discussion. In the DISCUSSION section, comparisons and references to studies by other authors should be added to initiate a scientific discussion.
2. Add research questions (the most important) in the Methodology section. In addition, it is worth adding a diagram - an illustration of the research process.
3. In the Conclusions section, add a description of the study's limitations (e.g. related to the fact that the sample of respondents was not representative).
Reviewer 2 Report
The article is well structured and gives very clear indications to political decision makers and professional associations about the gaps that need to be filled to fully integrate sustainable development into the interior design system.
In chapter 2 some concepts are redundant and are repeated, for example: lines 183-185, and lines 188-190. Lines 180-183 and lines 221-224 express a similar concept again. In lines 279, 308, in my opinion, the word "article" should be replaced with the word "paragraph".
In line 493 the "low" value must be changed to 2.09 to meet the threshold of the next value, which starts at 2.10. Please, check if it is a typo or if it is an error that excluded data.
Reviewer 3 Report
The subject matter explored within the manuscript is not only captivating but also carries significant meaning, making it a commendable endeavor. The manuscript demonstrates a commendable level of quality overall, and its chapter structure exhibits a robust foundation while maintaining a fluid and cohesive writing style.
Delving into the realm of interior design extends beyond the mere enhancement of living spaces; it profoundly impacts the lives of individuals and contributes to the propagation of sustainable development ideals. The insights presented in Chapter Four of the manuscript offer a glimpse into the authors' deep contemplation and thoughtful analysis of the subject matter. To enhance the depth of the discourse, it is advisable to expand further on how these innovative viewpoints could be practically realized.
Further refinement is needed in the research conclusion section. Specifically, the authors' assertion that this study introduces an all-encompassing new framework prompts important questions: How should this framework be practically applied? Are there any inherent limitations that need to be acknowledged? Once these aspects are thoughtfully addressed and fine-tuned, it would be appropriate to revise and update the article's abstract to accurately reflect the refined content.
It's essential to meticulously review and ensure the manuscript adheres to the proper formatting standards. This will contribute to the overall professionalism and readability of the manuscript, enabling it to effectively communicate its valuable insights to its intended audience.
It is necessary to check the manuscript in English and formatting.
Reviewer 4 Report
In general, I thought this was an interesting paper. However, there are many parts that repeat, sections are not well supported by references, the method could be better structured and explained, and the results need to be better presented. I think if these changes are considered then this would be a useful paper. The data collected is very useful. I have tried to include these comments and suggestions in my review.
1. The introduction section is poorly referenced and repeats itself in Section 2. I would recommend removing the introduction section and including more supportive references in the following section. For example, please include studies and data that support the increased importance of sustainable interior design. I think that sustainable interior design may refer to sustainable building design from the context of this paper.
2. It would also be helpful to include some examples of sustainable interior design. Lines 62-64 are not particularly useful. They include a wide variety of strategies. A relevant example could be the use of indoor fans that helped to reduce cooling energy consumption by 32% and maintained high human thermal comfort: Kent et al. Energy savings and thermal comfort in a zero energy office building with fans in Singapore.
3. Lines 73-76 repeat, but in more detail, the paragraph above. It might be worth merging these parts.
4. Lines 152-154: It might be more appropriate to explain the example of the 3Rs by using circular economy.
5. Section 2.2: It might be useful to emphasize whether Jordan or the Jordan BGC recommend any specific local or national standards for sustainability. Jordan GBC mentions LEED and BREEAM, which are likely the international versions of these certification systems, but I am unaware if any locally derived standards are advocated. The authors should also highlight that LEED and BREEAM most of the credits in these standards target building sustainability. Indoor environmental quality forms parts of the credits that are offered in both, yet the proportion of these is relatively small in comparison. This may be why IEQ satisfaction is similar across certified and non-certified BREEAM and LEED buildings: Altomonte et al. Indoor environmental quality and occupant satisfaction in green-certified buildings and Altomonte et al. Satisfaction with indoor environmental quality in BREEAM and non-BREEAM certified office buildings. I think that this, and other relevant literature, could be included.
6. Section 2.3.3: I would suggest that the authors review the WELL standard. This is more human centered than other certification labels, specifically designed to promote human health and well-being in buildings. This is also an international standard, which has seven certified buildings listed in its public directory from Jordan.
7. The research methodology (a) and (b) are confusing. If (a) was a comprehensive literature review, how is the primary data collection method a questionnaire? Does this mean there were two main sources of information that were gathered: Literature review and questionnaire? I think (a) could be omitted given that it is not included in the results.
8. Lines 351 to 365 should be added to Section 3.1 (e.g., “The final sample size included…) I would also recommend adding another section about the survey design. The authors only mention the Likert scale but not how many points, what questions were asked or how many. I think the scale was 5-points (from Figure 3), meaning that 0 was “disagree”, 5 was “agree”, and “3” was indifferent. Please clarify this. Also, how was Figure 2, awareness, gauged? Was this a simple yes/no question to five concepts?
9. Please improve Figure 3. The x-axis shows the questions, but they are incomplete. For example, the authors highlighted the last bar orange to highlight its importance. However, this says “Overall implantation of the…” All the text is required for the reader to understand the result. I would also recommend ordering the mean scores from lowest to highest, and more importantly, including the standard deviation. The graph shows the aggregate values and not the spread or distribution. Both are needed to understand the data.
10. The interpretation of the results is not clear. The authors say, “a mean score of 3.10, suggesting moderate adoption.” Without knowing the actual scale, the values become abstract. If a 5-point agree-disagree bipolar scale was used, then 3 would be “indifferent”. Therefore, a mean score of 3.10 cannot suggest moderate adoption. The authors need to indicate the mean score and its corresponding descriptor or meaning on the scale. An alternative solution would be to include the descriptors in Figure 3 – either as a secondary axis or next to the values on the primary axis.
11. Similar issues occur in Figure 4 and 5. Also, why is the formatting different from Figure 3? If the authors would like to retain the same formatting in Figure 4, please do not interpolate the points. The values are unrelated to each other.
12. I think that the results are interesting but do not really show what the authors want to reveal, or at least they are presented in a way that shows this. My main recommendation is to include clearly the 5-point scale and use this to indicate supportive or unsupportive evidence (e.g., above and below the neutral point), for each question.
Round 2
Reviewer 2 Report
Dear Authors in my opinion in the present form your article is much clearer and ready for publication. Even what you added in the discussion and in the conclusions to respond to the other reviewers made the work complete.
Reviewer 4 Report
I would like to thank the authors for addressing all my earlier comments. The revisions really helped to improve the manuscript. Some very minor changes could be considered (please find below).
1. Please join the text together in the abstract (unless there is a reason why they separated)
2. Parts that say "this paragraph" should probably read "this section"
3. Figure 2 and 3. The values above the boxplots can be rounded to integer values and "%" can be removed and added to the y-axis. This will prevent some values from being blocked
Author Response
Please, see the attachment.
